# Nanoporous Cauliflower-like Pd-Loaded Functionalized Carbon Nanotubes as an Enzyme-Free Electrocatalyst for Glucose Sensing at Neutral pH: Mechanism Study

**DOI:** 10.3390/s22072706

**Published:** 2022-04-01

**Authors:** Abdelghani Ghanam, Naoufel Haddour, Hasna Mohammadi, Aziz Amine, Andrei Sabac, François Buret

**Affiliations:** 1Chemical Analysis and Biosensors Group, Laboratory of Process Engineering and Environment, Faculty of Science and Techniques, Hassan II University of Casablanca, B.P 146, Mohammedia 20000, Morocco; abdelghani.ghanam@ec-lyon.fr (A.G.); hasna.mohammadi@fstm.ac.ma (H.M.); 2Univ Lyon, Ecole Centrale de Lyon, INSA Lyon, UCB Lyon 1, CNRS, Ampère UMR5005, 69130 Ecully, France; naoufel.haddour@ec-lyon.fr (N.H.); andrei.sabac@insa-lyon.fr (A.S.); francois.buret@ec-lyon.fr (F.B.)

**Keywords:** cauliflower-like Pd, enzyme-free electrocatalyst, glucose, mechanism, electrodeposition, functionalized carbon nanotubes

## Abstract

In this work, we propose a novel functionalized carbon nanotube (f-CNT) supporting nanoporous cauliflower-like Pd nanostructures (PdNS) as an enzyme-free interface for glucose electrooxidation reaction (GOR) in a neutral medium (pH 7.4). The novelty resides in preparing the PdNS/f-CNT biomimetic nanocatalyst using a cost-effective and straightforward method, which consists of drop-casting well-dispersed f-CNTs over the Screen-printed carbon electrode (SPCE) surface, followed by the electrodeposition of PdNS. Several parameters affecting the morphology, structure, and catalytic properties toward the GOR of the PdNS catalyst, such as the PdCl_2_ precursor concentration and electrodeposition conditions, were investigated during this work. The electrochemical behavior of the PdNS/f-CNT/SPCE toward GOR was investigated through Cyclic Voltammetry (CV), Linear Sweep Voltammetry (LSV), and amperometry. There was also a good correlation between the morphology, structure, and electrocatalytic activity of the PdNS electrocatalyst. Furthermore, the LSV response and potential-pH diagram for the palladium–water system have enabled the proposal for a mechanism of this GOR. The proposed mechanism would be beneficial, as the basis, to achieve the highest catalytic activity by selecting the suitable potential range. Under the optimal conditions, the PdNS/f-CNT/SPCE-based biomimetic sensor presented a wide linear range (1–41 mM) with a sensitivity of 9.3 µA cm^−2^ mM^−1^ and a detection limit of 95 µM (S/N = 3) toward glucose at a detection potential of +300 mV vs. a saturated calomel electrode. Furthermore, because of the fascinating features such as fast response, low cost, reusability, and poison-free characteristics, the as-proposed electrocatalyst could be of great interest in both detection systems (glucose sensors) and direct glucose fuel cells.

## 1. Introduction

Glucose electrooxidation reaction (GOR) has been extensively studied over the years for several applications such as in the development of glucose electrochemical sensors and biosensors for blood sugar monitoring [1,2,3] or the energy supply of bioelectronics-based devices by enzymatic and non-enzymatic glucose fuel cells [4,5,6]. GOR has also been involved in microbial fuel cell technology that converts glucose chemical energy to electrical energy [7]. As a reaction, GOR was first studied hundreds of years ago [8]; however, the last ten years have again attracted the scientific community’s interest due to its potential application in the aforementioned fields [9]. These applications have led to increasing demand for the development of inexpensive electrocatalysts that have high sensitivity, high selectivity, good stability, and low overpotential for glucose oxidation [10,11,12]. Although enzymes, such as glucose oxidase (GOx) and glucose dehydrogenase (GDH), have a high sensitivity and a great selectivity toward GOR, these biocatalysts suffer from degradation and instability, limiting their utilization in long-term applications [10,13,14]. In addition, their use in biosensing requires cumbersome immobilization procedures and a relatively high positive potential to oxidize the hydrogen peroxide (H_2_O_2_) produced by the enzymatic reaction [10,15]. In order to overcome these shortcomings, many efforts have been devoted to successfully developing non-enzymatic (abiotic) electrocatalysts using nanomaterials [3,10]. These nanomaterial-based enzyme mimetics, defined as “nanozymes’’, would become the next generation of artificial enzymes thanks to advantages such as high stability, low cost, high electrocatalytic activity, and excellent biocompatibility, as well as a simple and reproducible method for mass production [16,17]. 

Many studies have reported on the electrocatalytic activity toward the GOR of enzyme-free catalysts based on various materials such as noble metals [17,18], alloys [19,20], carbon-based materials [21], and other nanocomposites [22,23]. For example, Emir et al. documented a non-enzymatic glucose sensor based on a Ni-NPs/Ppy-modified graphite rod electrode for selective glucose sensing in alkaline media (0.1 M NaOH, pH 13.0) [24]. Huang et al. reported Pd-Co over a carbon nanotube-modified glassy carbon electrode (GCE) as a non-enzymatic catalyst for both glucose and H_2_O_2_ detection in alkaline conditions (0.1 M NaOH, pH 13.0) [25]. Shen et al. presented a novel platform based on a Pd-Au bimetallic cluster as an enzyme-free based glucose sensor in 0.1 M NaOH (pH 13.0) [26]. Moreover, Li et al. successfully synthesized MoS_2_ nanosheet-supported Au-Pd bimetallic nanoparticles (NPs) as peroxidase mimetics for H_2_O_2_ and glucose detection in aqueous NaOH solution (pH 13.0) [27]. Tang et al. reported a novel enzyme-free glucose sensor in 0.1 M NaOH based on Pd nanosheets deposited on Cu/Cu_2_O nanomaterials via galvanic replacement [28]. Additionally, electrocatalysts based on a carbon-supported Pd-Au binary composite for GOR in alkaline media (0.1 M NaOH) were synthesized via a pulse microwave-assisted polyol method [20,29]. However, the catalytic activity of most enzyme-like materials requires highly alkaline conditions, limiting their real application in some areas, such as Point-of-Care diagnosis, the on-site monitoring of glucose, and implantable non-enzymatic fuel cells.

More recently, some works reported the electrocatalytic activity toward GORs in neutral media of non-enzymatic nanomaterials based on Pt and Au [10]. Although Pt and Au nanomaterials showed high catalytic performance in a neutral buffered solution (pH 7.4) for GOR, their intrinsic properties were disrupted, as they were easily poisoned by intermediates and products generated during the GOR [30]. Furthermore, Pt and Au are very expensive compared to other materials and have a high overpotential for GOR [20]. Since catalytic sites can be blocked quickly by the chemisorption of glucose oxidation products under physiological conditions, the development of an efficient catalyst for GOR with good stability still remains a major challenge to be addressed. 

Pd-based nanomaterials are of great interest thanks to their excellent enzyme-like activity and good stability toward GOR. Moreover, Pd is at least 50 times more abundant in the earth’s crust than Pt and Au metals and its overpotential for GOR is lower [22]. Carbon-based nanomaterials such as carbon nanotubes (CNTs) [23,31], ordered mesoporous carbon (OMC) [32], graphene oxide (GO) [22], and reduced GO [33] have been investigated as carbon-based supporting materials in non-enzymatic sensor construction due to their excellent electrical conductivity. Several studies have described the electrocatalytic activity of carbon nanotubes (CNTs) modified with various Pd-based nanostructures. Indeed, the use of CNTs as carbon-based supporting materials has been an important strategy in the construction of non-enzymatic electrocatalysts, since it increases electroactive surface area for the immobilization of metallic nanomaterials, improving their chemical stability and electrical conductivity. Meng et al. reported the electrocatalytic activity toward the GOR of CNTs decorated with chemically synthesized Pd nanoparticles [34]. Ye et al. described the ability of chemically prepared Pd nanocubes to electrocatalyse glucose oxidation, showing great potential and stability for use as a catalyst in glucose sensors [35]. These studies suggest that the shape and size of the Pd particles have a significant effect on electrocatalytic activity. In comparison with the chemical methods for catalyst preparation, electrodeposition is a flexible technique allowing the rapid coating of conductive surfaces with thin films of catalysts of different sizes (micro- and nano-structures) and forms (wires, rods, tubes, ribbons, etc.) [36,37,38,39]. For instance, Jiang et al. reported the electrodeposition of cauliflower-like palladium nanoparticles used as catalysts for the reductive dechlorination of 4-chlorophenol in the field of pollutants removal [40]. To the best of our knowledge, the electrocatalytic activity of electrodeposited Pd for glucose electrooxidation has not been investigated to date, although this preparation method provides the opportunity to control the shape and size of deposited catalysts through electrolysis parameters (applied potential, the concentration of the metallic precursors, temperature, pH, duration of the electrolysis, etc.) [41].

In this study, SPCE surfaces modified with carboxylic acid f-CNTs were used as carbon-based supporting materials for the electrodeposition of PdNS catalysts. The electrocatalytic activity of the deposited PdNS catalysts was investigated for the direct glucose oxidation in a neutral buffered solution. The effect of the metallic precursor concentration and electrodeposition potential on the morphology evolution of electrodeposited Pd nanostructures were investigated using scanning electron microscopy (SEM). The electrocatalytic activity of PdNS catalysts toward GOR was characterized by Cyclic Voltammetry (CV) and Linear Sweep Voltammetry (LSV). A correlation between the morphology of PdNS catalysts and their electrocatalytic activity was observed. In addition, a mechanism for GOR was proposed based on the LSV response of electrocatalysts and the potential-pH diagram for the palladium–water system. Amperometry measurements were also carried out to assess analytical features of electrodeposited PdNS as non-enzymatic catalysts for glucose sensing in a neutral environment.

## 2. Materials and Methods

### 2.1. Chemicals and Reagents

Carboxylic acid-functionalized multi-walled carbon nanotube (f-CNT), >80% carbon basis, >8% carboxylic acid-functionalized, avg. D × L 9.5 nm × 1.5 µm, N,N-Dimethylformamide (DMF), Nafion (5 wt.%), D-(+)-glucose (≥99.5%), sulfuric acid (H_2_SO_4_, 98%), and perchloric acid (HClO_4_, 70.0–72.0%) were purchased from Sigma Aldrich (USA) and used as received without further purification. Palladium (II) chloride (PdCl_2_, 59.8 wt.%) was obtained from Johnson Matthey (UK). All other chemicals used were of analytical reagent grade. The electrochemical electrooxidation of glucose was performed in 0.1 M PBS (pH 7.4), prepared using analytical grade reagents, Na_2_HPO_4_ and KH_2_PO_4_ dissolved in distilled water, purchased from Solvachim (Casablanca, Morocco). An amount of 200 mM glucose stock solution was prepared in 0.1 M phosphate buffer solution (PBS) and stored at 4 °C. Glucose stock solution was allowed to mutarotate under magnetic stirring for at least 24 h before use.

### 2.2. Apparatus and Electrochemical Techniques

CV, LSV, and amperometric measurements were carried out using an electrochemical instrument, PalmSens BV (Houten, The Netherlands), connected to a computer and controlled by software named PSTrace5.8. A conventional three-electrode electrochemical system was used. The SPCE acted as a working electrode with a diameter of 3 mm. The reference electrode was a saturated calomel electrode (SCE) saturated with 3 M KCl, while a bare of stainless steel was used as a counter electrode. Electrode surface morphology was examined with scanning electron microscopy (SEM, JEOL model JSM-7401F, (JOEL, Croissy, France)).

### 2.3. SPCE Production

Screen-printed carbon electrodes (SPCEs) were home-produced with a 245 DEK (Weymouth, UK) screen-printing machine. Graphite-based ink (Electrodag 423 SS) from Acheson (Milan, Italy) was used for printing the working and counter electrodes. The substrate was a flexible polyester film (Autostat HT5) obtained from Autotype Italia (Milan, Italy). The homemade electrodes were produced in foils of 48 units. The diameter of the working electrode was 3 mm, resulting in a geometric area of 0.07 cm^2^. These electrodes were obtained as a gift from Prof. Fabiana Arduini at Tor Vergata University (Rome, Italy).

### 2.4. Preparation of f-CNT/SPCE

A dispersion of f-CNT was prepared by adding 10 mg of f-CNT powder into 10 mL of a dispersing agent (0.125% (*v*/*v*) Nafion in DMF) and sonicated for 3 h. A small volume (6 µL) of the dispersion was then drop-cast onto the working electrode surface of the SPCE in three steps of 2 µL. Afterward, the solvent was allowed to dry at 45 °C for 15 min, and an f-CNT “film” was left on the electrode surface (f-CNT/SPCE).

### 2.5. Electrodeposition of PdNS Catalyst

The Pd electrocatalyst was electrochemically deposited on the f-CNT/SPCE surface at a fixed deposition potential of −0.2 V vs. SCE for 300 s in a solution of 0.05 M HClO_4_ and 0.25 M H_2_SO_4_ containing the PdCl_2_ precursor, which resulted in the PdNS/f-CNT/SPCE sensor. The same procedure was followed to prepare the PdNS catalyst onto the SPCE surface without being previously modified with f-CNTs, resulting in the PdNS/SPCE sensor. A general schematic illustration of the preparation steps of the enzyme-free electrocatalyst-based glucose sensor is described in Figure 1.

### 2.6. Characterization of Electrocatalytic Activity for GOR

The electrocatalytic activity of electrodeposited catalysts was investigated by CV and LSV measurements at a well-defined potential window, with 8 mV as step potential and a scan rate of 10 mV s^−1^ in 0.1 M PBS at pH 7.4 containing an appropriate amount of glucose, as indicated in CV experiments and further electrochemical measurements. Chronoamperometry measurements were also carried out with a PdNS/f-CNT/SPCE sensor using amperometric batch analysis in 10 mL of a stirred (300 rpm) solution of 0.1 M PBS (pH 7.4) with a selected applied potential. When a stable baseline current was reached, the glucose was added successively, and the responses were recorded. All electrochemical measurements were performed at room temperature (25 °C) with three replicates. 

## 3. Results and Discussion 

### 3.1. Parameters Affecting the Electrocatalytic Activity of Pd/f-CNT toward Glucose in Neutral pH

#### 3.1.1. Effect of f-CNTs

The electrodeposition of PdNS on bare SPCE and f-CNT/SPCE was performed at −0.2 V vs. SCE for 1000 s in a solution of 0.05 M HClO_4_ and 0.25 M H_2_SO_4_ containing 1 mM Pd^2+^ (Appendix A). The CV characterization/polarization of electrodeposited PdNS in 0.5 M H_2_SO_4_ solution showed, for both electrodes, the typical electrochemical response of Pd surfaces. Indeed, CV curves exhibited anodic oxidation and cathodic reduction peaks corresponding respectively to Pd oxide formation during anodic scanning and a reduction in oxidized Pd in the reverse scan (Appendix A). It was observed that the oxidation and reduction peak currents increased with the presence of f-CNT on the electrode surface. This is due to a higher electroactive surface area of f-CNT/SPCE compared to the same geometric surface of bare SPCE. These results were consistent with previously reported studies [34,42].

The electrocatalytic behavior of PdNS-modified electrodes toward GOR was investigated using CV in 0.1 M PBS (pH 7.4), both in the absence (Figure 2A) and in the presence of 20 mM glucose (Figure 2B). In the glucose-free solution, an intense cathodic reduction peak was observed for both electrodes at around 0.0 V vs. SCE attributed to the reduction of palladium oxide species. The addition of glucose (20 mM) showed the anodic current peak of glucose oxidation at around 160 mV vs. SCE. This oxidative peak (*) appeared during the cathodic scan of both SPCE- and f-CNT/SPCE-based PdNS. This peak might be attributable to the oxidation of dehydrogenated D-glucose (previously adsorbed via dehydrogenation) or the oxidation of products adsorbed on the Pd catalyst during the direct scan (anodic scan) [27,43]. The current intensity of the anodic peak obtained with PdNS/f-CNT/SPCE was more than twice as high as that obtained with PdNS/f-SPCE, indicating that electrocatalytic activity of the PdNS catalyst toward GOR is greatly enhanced by the insertion of f-CNTs that synergize with the PdNS catalyst. Therefore, these f-CNTs were chosen for coating SPCE in the rest of this study.

#### 3.1.2. Effect of Electrodeposition Potential

The applied potential is an important factor in controlling the morphology and size of metallic electrodeposited particles [36]. Therefore, the electrodeposition of the PdNS catalyst on the f-CNT/SPCE surface was performed under different applied potentials (−0.2, 0.0, and +0.4 V vs. SCE) for 300 s in a solution containing 10 mM metallic precursor. The electrocatalytic performances of PdNS/f-CNT/SPCE toward 20 mM glucose were investigated in 0.1 M PBS by LSV. As indicated in Figure 3 (see also all voltammograms superimposed in Appendix A), decreasing the deposition potential to negative values, from +0.4 V to −0.2 V vs. SCE, contributed to an enhancement of electrocatalytic activity. The morphology of PdNS electrodeposited under different applied potentials was investigated by scanning electron microscopy. As shown in Figure 3, under +0.4 V vs. SCE, the electrodeposited Pd particles consisted of a smooth surface like a bulk Pd electrode. Terraces formed as massive microstructures condensed and agglomerated (Figure 3c). Furthermore, the morphology of the Pd changed slightly when the applied potential was +0.0 V, and the flower-like structures became denser and thicker with almost zero porosity, as illustrated in Figure 3b. However, decreasing the applied potential to −0.2 V resulted in the complete disappearance of the massive terraces, while a 3D cauliflower-like PdNS appeared on the electrode surface, as shown in Figure 3a. The cauliflower-like morphology of electrochemically synthesized Pd nanostructures is similar to that reported by Sun et al. for dichlorination catalysis [40]. These results indicated that deposition potential has a significant effect on the shape of Pd nanostructures, which subsequently improved the electrocatalytic activity toward GOR in a neutral buffered medium. 

These morphological studies demonstrate their complementarity with the electrochemical ones. The real surface area (RSA) of electrodeposited PdNS catalysts on electrodes, as a function of applied potential, was estimated by measuring the charge of the formation and reduction of PdO. This method was previously reported for evaluating the RSA of Pd electrodes and was based on the assumption that the charge density required to reduce a monolayer of PdO per unit of surface area was 0.424 mC cm^−2^ [44,45,46]. By following the procedure described in the Appendix A, the RSA was calculated for PdNS obtained at different applied potentials. The RSA of PdNS electrodeposited at −0.2 V vs. SCE (7.39 cm^2^) was higher than those at 0 V vs. SCE (4.66 cm^2^) and +0.4 V vs. SCE (0.52 cm^2^) (Figure 3). These values of RSA are consistent with the morphology evolution observed in the SEM images, as well as the electrocatalytic activity recorded by the LSV technique toward GOR. Therefore, −0.2 V vs. SCE was considered an optimal potential for the electrodeposition of PdNS on f-CNT/SPCE to enhance their catalytic activity by increasing the RSA in heterogeneous catalysis. This applied potential value was chosen for coating SPCE in the rest of this study.

#### 3.1.3. Effect of PdCl_2_ Precursor Concentration

The PdCl_2_ precursor solution concentration has a significant influence on the catalytic activities of metallic electrocatalysts [36]. Therefore, PdNS were electrodeposited at various PdCl_2_ precursor concentrations. Modified nanostructured sensors (PdNS/f-CNT) were then investigated for glucose electrooxidation.

As indicated in Figure 4A, SEM characterization was used to investigate the effect of metallic precursor concentration on the morphology evolution of electrodeposited Pd nanostructures. Indeed, the morphology of nanostructured Pd changed considerably with the precursor concentration during electrodeposition. Increasing the precursor concentration, the evolution of the Pd structure changed from microclusters to 3D cauliflower-like microstructured porous Pd. When the concentration of Pd^2+^ was 10 mM (Figure 4(a)), the morphology of Pd was cluster-shaped microstructures until they became bigger and bigger when 20 mM (Figure 4(b)) to 50 mM (Figure 4(c)) Pd^2+^ were used during the electrodeposition process. Up to 100 mM Pd^2+^ concentration of the precursor resulted in dense and large cauliflower-shaped Pd with a large electrocatalytic surface area on the f-CNT/SPCE surface, as shown in Figure 4d. These results demonstrate that the precursor solution concentration has a crucial effect on controlling the morphologies of Pd structures and supplying more catalytic sites to benefit the electrocatalytic oxidation of glucose molecules.

As shown in Figure 4B–E, the electrocatalytic activity of PdNS/f-CNT/SPCE toward GOR was investigated in 0.1 M PBS (pH 7.4) with (green line) and without 20 mM glucose (black line) using LSV measurements at a scan rate of 10 mV s^−1^. According to these results, the higher the concentration of the precursor, the more the enhanced electrocatalytic activity of PdNS/f-CNT/SPCE toward GOR will be. In addition, a shift of the glucose oxidation peak toward positive values of potentials from 165 to 300 mV vs. SCE was observed with the increase in precursor concentration (10 to 100 mM Pd^2+^). 

In addition, amperometric measurements on PdNS/f-CNT/SPCE were performed at +0.3 V vs. SCE. The results obtained are consistent with those obtained by LSV. This method enables a choice according to the best results in terms of detection limit and sensitivity. Figure 4F shows amperometric responses obtained from PdNS/f-CNT/SPCE at an applied potential of +0.3 V vs. SCE after a successive glucose addition in the range of 1–20 mM.

The sensitivity and detection limit (DL) were significantly improved when increasing the precursor concentration. Table 1 summarizes the analytical parameters provided by amperometry. On the basis of these results, a 100 mM precursor concentration was chosen as the optimal concentration, providing 3D porous cauliflower-like Pd nanostructures with high sensitivity and a low DL.

### 3.2. Mechanism Proposed for Glucose Oxidation on PdNS/f-CNT Biomimetic Nanocatalyst in Neutral pH

It should be noted that in all LSV voltammograms (black curves in Figure 4), two peaks, O1 and O2, appeared in the glucose-free background PBS solution. As observed, the current intensity of peak O1 increases with the concentration of the precursor up to 50 mM. This peak was only slight in the structure of the electrocatalyst obtained with 100 mM. To better understand the mechanism underlying these peaks, the Pourbaix diagram (E-pH) of the Pd–water system was used (Appendix A). The theoretical potential values were calculated by replacing the pH with a 7.4 value in all linear equations (E-pH) of the Pourbaix diagram [47], as described in detail in the Appendix A. Therefore, the obtained values were compared with those corresponding to the surface oxidation peaks (O1 and O2) of the Pd electrocatalyst in the glucose-free background solution. 

Based on the potential values representing the O1 and O2 peaks in the LSVs (Figure 4, black curve), it can be assumed that these two peaks correspond to oxidation equations, Equations (1) and (2), of the catalyst surface in the background solution. However, the O3 peak corresponds to Equation (3), which represents the balanced equation of glucose oxidation at the Pd catalyst, involving glucose molecules and PdO oxidizing species.
Pd_2_H ⇄ 2Pd + H^+^ + e^−^(1)
Pd + H_2_O ⇄ PdO + 2H^+^ + 2e^−^(2)
PdO + C_6_H_12_O_6_ (glucose) → Pd + C_6_H_10_O_6_ (gluconolactone) + H_2_O(3)

Pd catalysts are well known for their ability to adsorb and absorb H, forming the compact and stable surfaces of Pd_2_H hydrides. Therefore, the peak (O1) that appeared in the potential range of −0.6 to −0.3 V vs. SCE (hydrogen region) can be attributed to the hydrogen desorption from the Pd catalyst, while the oxidation peak O2 (oxygen region) can be due to the electrooxidation of dehydrogenated Pd (Figure 1A). 

In addition, the evolution of hydrogen was significantly remarked on in the catalyst structures from Figure 4(a–c). Indeed, the oxidation peak (O1) of Pd_2_H in the background solution increases with a precursor concentration up to 50 mM, while this peak is very weak at 100 mM, indicating that this catalytic structure (Figure 4(d)) has low proton adsorption. H_2_ interaction with Pd-based catalysts depends on the structure of the materials by influencing adsorbate binding energies [48]. The electrodeposition of PdNS catalysts with a precursor concentration of 100 mM probably induces structural changes. For example, Sakamoto et al. demonstrated that two phases of adsorbed hydrogen can exist in Pd-based alloys with elements that do not adsorb hydrogen under normal conditions [49,50]. 

After adding 20 mM glucose to the supporting electrolyte, both the O1 and O2 peaks increased (Figure 4). For the O1 peak, this may be due to both the desorption of H (H coming from the medium) from the Pd catalyst and the dehydrogenation of glucose molecules via their adsorption onto the Pd catalyst [27]. This result is consistent with Pletcher’s theory for concentric adsorption, which involves hydrogen extraction followed by the simultaneous adsorption of the organic species on the metallic catalyst, as shown in Figure 1B [51]. At 100 mM Pd^2+^, the O1 peak is dramatically increased, which may be due to the strong dehydrogenation of glucose molecules by the catalyst, leading to the formation of Pd_2_H on this cauliflower-like structure. These results indicated that the PdNS structure obtained with 100 mM of the precursor was not favorable to hydrogen adsorption in neutral conditions, while it was supportive of glucose dehydrogenation. The significant enhancement of the oxidation peak of dehydrogenated Pd (O3) is probably due to the electrocatalytic oxidation of glucose molecules already adsorbed and/or accumulated on the surface of the Pd catalyst (Figure 1B). Furthermore, the electrocatalyst response of PdNS showed satisfactory results in terms of stability in GOR, demonstrated by their regeneration after each scan. Assuming that glucose molecules can be oxidized to gluconolactone in neutral conditions by a two-electron electrochemical reaction [27,52] and that PdO may act as an oxidizing species, a possible electrocatalytic mechanism on the PdNS/f-CNT/SPCE surface toward GOR was proposed, as indicated in Figure 5.

In the proposed mechanism of glucose oxidation on the PdNS catalyst in neutral media, the peak O2 corresponds to oxide formation (PdO) generated by the Pd from the hydrogen desorption reaction (O1). In addition, the peak O3 is attributed to glucose oxidation, which is induced with high-current intensity, as the current intensity of O1 increases.

### 3.3. Performance of PdNS/f-CNT-Based Glucose Sensors

#### 3.3.1. Performance Comparison of PdNS/f-CNT as Non-Enzymatic Glucose Sensors in Neutral pH

Under the optimal conditions, the PdNS/f-CNT/SPCE sensor at the fixed potential of +0.3 V vs. SCE was successfully used for amperometric glucose detection at neutral pH. The response current of the non-enzymatic sensor showed a wide linear regression against a glucose concentration in the range of 1–41 mM with a DL of 95 µM (S/N = 3) and sensitivity of 9.26 µA mM^−1^ cm^−2^. Importantly, this range covers the normal human blood glucose level. Table 2 summarizes the analytical performances of the newly designed, non-enzymatic glucose sensor with those previously conducted in neutral environments. As a result, the analytical features of the proposed PdNS/f-CNT catalyst are comparable to those described in the literature. Furthermore, our designed non-enzymatic sensor can be utilized for practical sample testing with favorable accuracy and precision. 

#### 3.3.2. Selectivity of PdNS/f-CNT-Based Glucose Sensors

As mentioned previously, one of the main challenges in the non-enzymatic sensing of glucose is the electrochemical signals caused by some interfering substances such as ascorbic acid (AA), paracetamol (PCM), dopamine (DA), and galactose (Gal). Although the selectivity of non-enzymatic glucose sensors is often lower than their enzymatic counterparts, our developed glucose sensor still reveals a relatively good selectivity against various interfering species, such as fructose and sucrose. Amperometric measurements were conducted by applying +0.3 V vs. SCE to PdNS/f-CNT/SPCE in 0.1 M PBS (pH 7.4) containing glucose and interfering species. Table 3 summarizes the electrochemical responses of all tested molecules in the presence of glucose. No amperometric response could be identified for Suc and Fru, even at high concentrations of 1 and 10 mM. However, Gal, DA, PCM, and AA could cause interference with the glucose oxidation signal. Further discrimination of electroactive species on the PdNS/f-CNT/SPCE non-enzymatic sensor can be accomplished by: (i) using ion-exchange membranes over the sensor (e.g., if charged interference species are present), (ii) coupling the sensor with a proper sample preparation procedure to select the target analyte, and so on.

## 4. Conclusions

In conclusion, this study proposed a simple and cost-effective electrodeposition method to synthesize porous cauliflower-like Pd nanostructures (PdNS) on f-CNT/SPCE as an enzyme-free sensor for direct glucose detection in a neutral buffered solution (pH 7.4). Thanks to its high electrical conductivity, f-CNT increased the electrochemical surface area of the bare electrode, which promoted the electrodeposition of an important amount of particles of the PdNS catalyst. Furthermore, f-CNTs were used as carbon-based supporting materials for the construction of the PdNS electrocatalyst. Indeed, the electrocatalytic activity of the PdNS catalyst toward glucose oxidation was greatly enhanced by the insertion of f-CNTs that synergize with the PdNS catalyst. The PdCl_2_ metallic precursor concentration and the electrodeposition conditions were studied and discussed. SEM analysis confirmed that these parameters had a significant effect on the morphological structure evolution of the deposited Pd nanostructures. Besides, a good correlation between the morphology evolution, structure, and electrocatalytic activity of the electrodeposited PdNS electrocatalyst toward glucose was observed. Furthermore, glucose electrochemical behavior and the potential-pH diagram for the palladium–water system enabled us to propose a mechanism for the glucose electrooxidation reaction. The proposed mechanism would be beneficial, as the basis, to achieve the electrocatalyst’s highest activity by selecting the suitable potential range.

Nevertheless, the use of this new generation of abiotic sensors, for real applications, requires the acceptable selectivity of the f-CNT/SPCE-based PdNS, which might be obtained by:Reducing the applied potential by inserting non-precious metals, polymers, etc.;The use of ion-exchange membranes over the sensor (e.g., electrostatic repulsion if charged interference species are present);Coupling the non-enzymatic sensor with a sample preparation procedure to select the target analyte.

## Data Availability

Not applicable.

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
