# Peer review of "Nanoporous Cauliflower-like Pd-Loaded Functionalized Carbon Nanotubes as an Enzyme-Free Electrocatalyst for Glucose Sensing at Neutral pH: Mechanism Study"

_sensors, 2022, doi:10.3390/s22072706_

Round 1
Reviewer 1 Report
In this manuscript, the authors prepared Pd nanostructures modified CNTs for glucose sensing and studied the mechanism of the glucose electrooxidation reaction on the Pd-CNT-electrode. It sounds okay for the mechanism study. However, as a glucose sensor, I do not see significant improvements of its performance compared to other reported electrochemical glucose sensors, as in linear range, sensitivity, detection limit. For the selectivity study, the responses from DA/PCM/AA are much larger than glucose itself. I would suggest the authors to further improve the performance of the proposed sensor, and focus more on the mechanism studies.
Reviewer 2 Report
The article presents a thorough study of the use of Pd nanocomposites to demonstrate their utility in the direct electrooxidation of glucose at neutral pH with nice explanations at the fundamental level. I recommend the paper for publication subject to some comments below.
- A CV of the PdCl2 would be useful in rationalising the selection of the electrodeposition potential which plays a key role in controlling the shape and size of the nanomaterials.
- Regarding the electroactive surface area measurements on page 6 the value of 0.424 mC cm-2 is used in the text line 268 while section 1.1 of the supplemental information gives a value of 424 C cm-2. The Qo is given as S/V where the units of this term would be C V-1 s in the RSA expression and so how do these units cancel to give RSA in cm-2? Shouldn’t Qo be measured in coulombs?
- Figure 5 (which should be labelled a Scheme) shows the proposed mechanism – the interaction between surface Pt and the C1 OH group (hemiacetalic and therefore preferentially dehydrogenated?) should be clarified in the drawing (B))
- The galactose interference is unsurprising given the structural similarity at C1 - this would be an issue for monosaccharide differentiation (assuming the more charged redox active species can be eliminated). Some further explanation of how to further discriminate would be useful.
- Table 2 should refer to the article below which reports bimetallic and trimetallic (including Pd) reports of glucose nonenzymatic sensing at neutral pH.
Baljit Singh et al ‘PtAuPd decorated carbon nanochips and nanotubes for direct glucose sensing - role of support material and efficient Pt utilisation’, Sensors & Actuators: B. Volume 205, 15 December 2014, Pages 401–410.
Reviewer 3 Report
Please see the attachment.

Reviewer 4 Report
Review on a manuscript entitled “Nanoporous Cauliflower-like Pd-Loaded Functionalized Carbon Nanotubes as an Enzyme-free Electrocatalyst for Glucose Sensing at Neutral pH: Mechanism Study”
Authors in this study discuss the development of a novel functionalized carbon nanotube (f-CNT) supporting nanoporous cauliflower-like Pd nanostructures (PdNS) as an enzyme-free interface to glucose Electrooxidation Reaction (GOR) at a neutral medium (pH 7.4).
The draft is interesting and well organized. Only very minor improvements and corrections should be made to the manuscript.
Comment for the authors:
Recent papers on the development of non-enzymatic sensors (Amperometric Nonenzymatic Glucose Biosensor based on Graphite Rod Electrode Modified by Ni-nanoparticle/Polypyrrole Composite. Microchemical Journal 2021, 161, 105751.) and reviews on glucose sensors (Charge transfer and biocompatibility aspects in conducting polymers based enzymatic biosensors and biofuel cells. Nanomaterials 2021, 11, 371.) should be overviewed and discussed.
The typing error in line 39: remove * from ha*s.
The font size in figure 4 is too small.
‘Figure 5. Schematic representation of proposed mechanism’ should be explained more clearly and more in detail.
Round 2
Reviewer 1 Report
Thanks for the authors' response and revision. The manuscript is acceptable now.